# Kinetic Variables as Indicators of Lower Limb Indirect Injury Risk in Professional Soccer: A Systematic Review

**DOI:** 10.3390/jfmk10020228

**Published:** 2025-06-16

**Authors:** Jorge Pérez-Contreras, Juan Francisco Loro-Ferrer, Felipe Inostroza-Ríos, Pablo Merino-Muñoz, Alejandro Bustamante Garrido, Felipe Hermosilla-Palma, Ciro José Brito, Guillermo Cortés-Roco, David Arriagada Tarifeño, Fernando Muñoz-Hinrichsen, Esteban Aedo-Muñoz

**Affiliations:** 1Escuela de Doctorado de La Universidad de Las Palmas de Gran Canaria (EDULPGC), 35016 Las Palmas, Spain; jperez51@santotomas.cl; 2Escuela de Ciencias del Deporte y Actividad Física, Facultad de Salud, Universidad Santo Tomas, Santiago 8370003, Chile; felipe.inostroza.311@gmail.com (F.I.-R.); alejandrobustamanteg@gmail.com (A.B.G.); 3Departamento de Ciencias Clínicas, Universidad de Las Palmas de Gran Canaria, 35016 Las Palmas, Spain; juanfrancisco.loro@ulpgc.es; 4Departamento de Educação Física, Instituto de Ciências da Vida, Universidade Federal de Juiz de Fora, Governador Valadares 35010-180, Brazil; ciro.brito@ufjf.br; 5Núcleo de Investigación en Ciencias de la Motricidad Humana, Universidad Adventista de Chile, Chillán 3780000, Chile; pablo.merino@usach.cl; 6Programa de Engenharia Biomédica, Instituto Alberto Luiz Coimbra de Pós-Graduação e Pesquisa de Engenharia (COPPE), Universidade Federal do Rio de Janeiro, Rio de Janeiro 21941-853, Brazil; 7Departamento de Educación Física, Deportes y Recreación, Facultad de Artes y Educación Física, Universidad Metropolitana de Ciencias de la Educación, Santiago 7760197, Chile; 8Pedagogía en Educación Física, Facultad de Educación, Universidad Autónoma de Chile, Talca 3460000, Chile; felipe.hermosilla@uautonoma.cl; 9Escuela de Ciencias de la Actividad Física, El Deporte y la Salud, Facultad de Ciencias Médicas, Universidad de Santiago de Chile, Santiago 8370003, Chile; 10Faculty of Life Sciences, Sports Coach, Universidad Viña del Mar, Viña del Mar 2520000, Chile; guillermo.cortes@uvm.cl; 11Laboratory of Applied Neuromechanics, School of Kinesiology, Faculty of Medical Sciences, Universidad de Santiago de Chile (USACH), Santiago 9170124, Chile; david.arriagada.t@gmail.com; 12Laboratorio de Actividad Física, Salud y Rendimiento Humano, Departamento de Kinesiología, Universidad Metropolitana de Ciencias de la Educación, Santiago 7501173, Chile; fernando.munoz_h@umce.cl

**Keywords:** football, injury incidence, muscle strength, dynamometry

## Abstract

**Background:** The high demands of professional soccer predispose players to musculoskeletal injuries. The primary challenge for identifying potential risk factors lies in determining the appropriate assessment methods and indicators to consider. Kinetic variables have been identified as potential indicators of injury risk. **Objectives:** To conduct a systematic review of the literature analyzing the relationship between kinetic variables and the risk of indirect musculoskeletal injuries of the lower limb in professional soccer players. **Methods:** A search was conducted on Web of Science, PubMed, and Scopus following the PRISMA guidelines. The search included articles that link kinetic variables assessed through dynamometry to indirect lower limb injuries. Sample characteristics, assessments, injury follow-ups, and statistical results were extracted for qualitative synthesis. **Results:** A total of 1096 studies were initially identified, of which 380 duplicates were removed. After screening 716 articles by title and abstract, 631 were excluded. Subsequently, 85 full-text articles were examined, resulting in 11 studies being included. Of the included articles, 10 used isokinetic dynamometry and the Nordic hamstring curl test to assess lower limb strength. **Conclusions:** The results of this review indicate that kinetic variables, particularly isokinetic strength measures at different angular velocities, are consistently associated with indirect musculoskeletal injury risk in professional soccer players. The most relevant indicators include eccentric hamstring force and concentric quadriceps torque, which help identify strength deficits and muscular disequilibrium. Monitoring these variables through validated assessments enables the development of targeted prevention strategies. Additionally, injury risk assessment should integrate kinetic data with contextual indicators such as well-being, fatigue, and training load.

## 1. Introduction

Soccer is an intermittent sport that requires physical and technical efforts at variable intensities over the course of matches. Explosive actions require an optimal physical condition to face competitive demands [1,2,3]. However, the high demands of soccer also increase injury risk, which is influenced both by external factors (number of matches, climate, type of competition) and internal factors (physical condition, health, and training load) [4,5,6]. An injury is defined as any physical trauma sustained by a player as a result of a soccer match or training session, regardless of whether the player requires medical care or must interrupt his training regime [7]. The absence of players in competitions due to injury not only affects team performance [8] but also the financial resources of clubs [9,10]; therefore, injury prevention becomes key in sports environments.

Muscle injuries can be classified as direct and indirect. The first is caused by external impact—e.g., contusions and lacerations—while the latter affects the musculoskeletal system either functionally or structurally [11]. Some functional injuries result from overexertion—e.g., type 1A muscle fatigue or type 1B delayed onset muscle soreness—whereas others are associated with neuromuscular dysfunction that may be caused by spinal disorders or abnormalities (type 2A) or muscle-specific conditions (type 2B). Conversely, structural injuries involve damage to tissue and can be categorized into partial tears (type 3A and 3B) or full tears and tendon avulsions (type 4), which can greatly impair muscle function. This classification aids in the diagnosis, treatment, and prevention of sports injuries [12,13].

Several authors have analyzed the epidemiology of sports injuries in different leagues, genders, age groups, and league divisions [4,8,14,15]. In professional soccer, muscle injuries have been found to account for one-third of injuries in high-performance athletes; of these, 92% are indirect injuries that affect the main four muscle groups of the lower limbs, these being the hamstrings (37%), adductors (23%), quadriceps (19%), and calf muscles (13%) [16]. In addition, Waldén et al. [17] found that 14% of injuries during a season in a professional soccer team occurred in the inguinal and hip areas, and 63% were related to the adductor muscles. A higher incidence of muscle injuries has also been reported during periods of increased match frequency (or congested schedules) [4,18]. One of the main challenges in identifying risk factors for injuries is selecting appropriate tests and indicators for assessment [19]. The evidence suggests that some factors—both modifiable and non-modifiable—can predispose athletes to injuries [20]. Some of the most important indicators include low levels of strength (weakness), fatigue, and asymmetries, which have been widely researched in relation to the incidence of musculoskeletal injuries [21,22,23]. However, there is no consensus on which variables are the most representative for identifying injury risk and how they should be interpreted in practice.

In this context, Pedley et al. [24] conducted a review that revealed that 79% of the studies using drop jump tests identified associations with future injuries, whereas only 8% of the studies with countermovement jump (CMJ) showed a similar relationship. Meanwhile, 57% of unilateral assessments were associated with injury risk, whereas performance-based tests (jump height or distance) showed a weaker association with 30% of cases. Furthermore, assessments that analyzed kinetic and kinematic variables presented a relationship with injuries in 89% of studies [24]. These findings underscore the importance of including advanced biomechanical measures for a more accurate assessment of injury risk.

A key factor for injury prevention is the presence of muscular asymmetries, especially those derived from previous injuries. It has been demonstrated that asymmetries ≥15% in the hamstrings eccentric force and ≥10% in the quadriceps concentric torque significantly increase the risk of muscle injury [25,26]. In addition, kinetic variables during the landing phase of drop jump tests more accurately reflect the injury mechanism, highlighting the importance of assessing not only the magnitude but also the direction and evolution of asymmetries throughout the rehabilitation process [27].

Strength measures have been established as a key tool for injury prevention and sports readaptation [28,29]. However, the selection of the method for calculating the asymmetry remains a point of debate, as different formulas can yield variable results [27]. Recent research suggests that dominant and non-dominant limbs may exhibit different asymmetry levels in jump tests, emphasizing the need for further exploration of their impact on injury risk in sports [30,31]. In this sense, using more standardized methodologies and integrated assessment tools will enhance the identification of players at risk while supporting the development of more effective preventive strategies.

Although kinetic variables are frequently used in performance assessments, their utility in predicting non-contact injuries, particularly in professional soccer, remains unclear. This knowledge gap highlights the need to better understand their role in injury surveillance and prevention.

Given the relevance of kinetic variables in injury prevention and performance monitoring, this systematic review aims to analyze the relationship between kinetic variables and the risk of non-contact musculoskeletal injuries of the lower limb in professional soccer players. By synthesizing the available evidence, this study seeks to identify trends, methodological limitations, and knowledge gaps that may inform injury prevention strategies in this specific context. This review specifically aimed to systematically examine the role of kinetic variables in predicting non-contact lower limb injuries in professional soccer.

## 2. Materials and Methods

The study was conducted following the Preferred Reporting Items for Systematic Reviews and Meta-Analyses (PRISMA) guidelines [32]. The review protocol was registered in International Prospective Register of Systematic Reviews (PROSPERO) website on 20 May 2024 (ID 538044).

### 2.1. Eligibility and Exclusion Criteria

The articles included in the review were selected following the PICO (Population, Intervention, Comparison, and Outcomes) methodology. The inclusion and exclusion criteria are detailed in Table 1.

### 2.2. Electronic Data Search

The systematic search was conducted in the following databases: Web of Science, PubMed and Scopus, including articles published between 1 January 2010 and 14 November 2023. The period selected allowed the analysis to focus on more recent and potentially relevant evidence, primarily due to the accelerated development of accessibility to high-quality technology in the context of professional soccer. To access these databases, the institutional credentials of Universidad de Las Palmas de Gran Canaria, Spain, were used.

The search strategy was a combination of controlled vocabulary (MeSH), specific free-text terms, intersection Boolean operators (AND), and union (OR) and truncation (*) Boolean operators. The search equation for WOS and Scopus was as follows: (“soccer” OR “football*”) AND (“Kinetic*” OR “strength*” OR “force*” OR “power*” OR “rate of force development” OR “imbalance*” OR “asymmetr*”) AND ((“Ruptur*” OR “Strain*” OR “tear*” OR “fatigue”) AND (“Lower Extremit*” OR “Lower Limb*” OR “Quadricep*” OR “Hamstring*” OR “adductor*” OR “tendon”)) AND (“Risk Factor*” OR “relation*” OR “associat*” OR “Correlation*”). For PubMed, the following search equation was used: (“soccer” OR “football*”) AND (“Kinetic*” OR “strength*” OR “force*” OR “power*” OR “rate of force development” OR “imbalance*” OR “asymmetr*”) AND ((“Ruptur*” OR “Strain*” OR “tear*” OR “fatigue”) AND (“Lower Extremity”[Mesh] OR “Lower Extremit*” OR “Lower Limb*” OR “Quadricep*” OR “Hamstring*” OR “adductor*” OR “tendon”)) AND (“Risk Factors”[Mesh] OR “Risk Factor*” OR “relation*” OR “associat*” OR “Correlation*”)

### 2.3. Study Selection

To process the results, the guidelines proposed by Rico-González et al. [34] were followed. A CSV was downloaded from each database to then copy the data onto an Excel spreadsheet. A coding system was employed to select the articles, in which 1 = included, 2 = duplicated, and 3 = excluded. The elimination of duplicated articles and initial coding was conducted separately by two reviewers (J.P.-C. and F.I.-R.), who analyzed titles and abstracts. Once the process was completed, the results were compared, and studies were selected for the following stage.

### 2.4. Data Collection Process

Three reviewers (J.P.-C., F.I.-R., and P.M.-M.) independently recorded the characteristics of the studies in an Excel spreadsheet, including details about samples, follow-up of musculoskeletal injuries, measured kinetic variables, assessment instruments, and results.

### 2.5. Risk of Bias

Two reviewers (F.I.-R. and P.M.-M.) independently assessed the risk of bias in the studies; when there were discrepancies, a third reviewer (J.P.-C) was consulted and resolved the disagreements. For this purpose, the Downs and Black checklist [35] was employed, of which only item 27 (power) was modified by assigning 1 point if a power analysis had been conducted and 0 points if this had not been conducted or was deemed inappropriate. Based on their scores, the studies were classified into the following categories [36]: excellent (26–28), good (20–25), fair (15–19), and poor (14 or below). To measure the level of agreement among the reviewers, Kohen’s kappa was analyzed, considering the categories of agreement (>0.81) almost perfect; (0.61 to 0.80), substantial; (0.41 to 0.60) moderate; (0.21 to 0.40) regular, (0.01 to 0.20) slight, and (<0.00) poor [37].

## 3. Results

### 3.1. Search Results

The initial search yielded 1096 results, of which 380 duplicated items were removed, with a total of 716 potentially relevant articles. During the initial coding, 631 articles were excluded upon the analysis of their title and abstracts. The full text of the 85 remaining articles was inspected based on the inclusion and exclusion criteria. Eleven articles were included in the review [25,26,38,39,40,41,42,43,44,45,46]. The process is presented in Figure 1.

### 3.2. Characteristics of the Studies

The characteristics of the population and indirect injury follow-up are summarized in Table 2 and Table 3, respectively. Of the 11 studies included, 10 have a prospective cohort design, and only 1 is retrospective [40]. All the studies included male soccer players, as no studies with a female population were found. Considering all the articles included, 2403 individuals were analyzed, of which 517 presented indirect injuries of the lower limb, and 1886 had no injuries. All the studies recorded indirect injuries to the hamstrings. Regarding other affected muscles, Fousekis et al. [25] and Liporaci et al. [26] reported on quadriceps on rectus femoris and hip adductor muscle injuries, respectively. The minimum follow-up time was one season [25,26,38,39,40,41,46], but some studies covered two [43,44] and others four [42,45].

### 3.3. Results of Risk of Bias

Regarding the methodology quality assessment conducted using the Downs and Black checklist for the included studies, reviewer agreement was substantial (Cohen’s Kappa = 0.673). Two articles obtained the score required to be classified as “good” [25,42], while all the other studies were categorized as “fair”, as shown in Table 4. Items 8, 13, 14, 15, 23, and 24 showed the most notable methodological shortcomings, mainly attributable to the observational design of the included studies and to a notable absence of experimental studies.

### 3.4. Strength Tests

Ten of the articles included used isokinetic dynamometry to assess lower limb strength [25,26,38,39,40,41,43,44,45,46], and two employed the Nordic hamstring curl test [42,44]. Detailed information about these tests—e.g., analyzed variables and outcomes—is presented in Table 5 and Table 6 for studies that included isokinetic dynamometry and the Nordic hamstring curl, respectively.

#### 3.4.1. Strength Assessment Using Isokinetic Dynamometer

The isokinetic evaluation protocols involved both concentric and eccentric testing for knee flexors but only concentric protocols for knee extensors. Repetition angular velocities ranged from 60°/s to 300°/s for concentric contractions of both knee flexors and extensors. For eccentric knee flexor contractions, velocities ranged from 30°/s to 180°/s. While peak isokinetic torque was the primary outcome variable, its behavior was analyzed across multiple dimensions. A glossary detailing these dimensions is provided in Table 5.

#### 3.4.2. Strength Assessment Using the Nordic Hamstring Curl Test

Two distinct evaluation protocols were identified for the Nordic hamstring flexion test: eccentric and isometric. The eccentric protocol, consistently applied across both studies [42,44], required participants to kneel on a padded board and descend in a controlled manner, striving to maintain a neutral trunk and hip position. Conversely, the isometric protocol utilized the same kneeling position with the knees extended, where participants performed a maximal isometric contraction for 5 s [42].

The primary findings from these studies focused on eccentric force production and eccentric torque [42]. Specifically, an absolute eccentric force below 337 N and a relative eccentric force below 4.35 N/kg were associated with relative risks of 4.4 and 2.5, respectively, for future injury. Similarly, an eccentric torque threshold of 145 N·m and a relative torque threshold of 1.86 N·m corresponded to relative risks of 3.6 and 2.9, respectively. Isometric strength variables [42] and maximum strength imbalance [44] between limbs did not show significant results, as shown in Table 6 and Table 7.

## 4. Discussion

Optimal physical fitness is key to facing the demands of professional soccer, as it also plays a crucial role in injury risk prevention, especially muscle injuries, which account for one-third of injuries in this sport [16]. Internal factors such as strength, fatigue, and asymmetries, together with kinetic variables associated with neuromuscular fatigue, have been proposed as risk indicators, allowing for the determination of thresholds that link physical performance to injury probability [52,53]. The purpose of this study was to conduct a systematic review of the literature to analyze the relationship between kinetic variables and the risk of indirect musculoskeletal injuries of the lower limb in professional soccer.

### 4.1. Strength Assessment Using Isokinetic Dynamometer

The results show a wide range of torque variables (type of contraction and execution speed) generated by the knee flexor–extensor musculature as possible risk factors. According to van Dyk et al. [45], a higher risk rate for tear injuries is associated with a deficit in the maximum torque production of the quadriceps during concentric movements at 60°/s, normalized by body mass (N·m/kg) (OR = 1.41; IC 95% 1.03–1.92; *p* = 0.03), as well as a deficit in the maximum eccentric torque production of the hamstring at 60°/s, normalized by body mass (N·m/kg) (OR = 1.37; IC 95% 1.01–1.85; *p* = 0.04). Nevertheless, both variables have lower probabilities as predictors of hamstring tears. Another study using a multiple linear regression model discovered that the concentric torque at 240°/s of the dominant leg’s hamstrings influenced the hamstring injury rate (β = 0.01 ± 0.01) [46]. However, the clinical magnitude is considerably small compared to other factors in the model, such as age, hamstring injury history, and the results of the Nordic hamstring curl test.

An interesting analysis involves determining cut-off points to classify dichotomous variables as risk factors. When analyzing the concentric torque of the quadriceps at 300°/s, van Dyk et al. [44] found that an increase of one standard deviation above the mean in the maximum concentric torque of the quadriceps, normalized by body mass, increased the risk of hamstring tears (HR = 2.06; 95% CI 1.21–3.51), with a small to moderate prevalence [54]. In turn, Lee et al. [41] discovered that values below 2.4 N·m/kg for the hamstring eccentric torque peak at 30°/s indicate a large effect size for the risk of tears (Adjusted OR = 5.59; IC 95% 2.20–12.92; *p* < 0.001). Finally, van Dyk et al. [43] studied the rate of torque development (RTD) at 30 ms, 50 ms, and 100 ms without significant findings. These results are likely influenced by the lower reliability of early RTD (0 to 100 ms) compared to the late RTD of the knee extensors (150 to 250 ms) [55]; however, the authors did not report the reliability of the early RTD of the knee flexor–extensors. Therefore, early RTD may preliminarily be an appropriate risk factor for hamstring tears; nevertheless, further research is required on late RTD, as this has demonstrated more reliability between sessions.

### 4.2. Ratios and Asymmetries Derived from Isokinetic Strength Assessments

Using ratios in the assessment of isokinetic strength is a common practice. Although the hamstring/quadriceps (H:Q) ratio is the most popular, the hamstring/hamstring (H:H) ratio and quadriceps/quadriceps (Q:Q) ratio were also employed. To calculate them, different torque combinations are used, considering contraction type, execution speed and limb dominance. The most critical findings are related to concentric H:Q at 60°/s as a possible risk factor for hamstrings. Lee et al. [41] found that the absolute value of concentric H:Q at 60°/s could be considered a protective factor for hamstring tears; however, when that same variable is conditioned to a risk threshold, if the concentric H:Q at 60°/s is under 50.5%, it then becomes a risk factor for this phenomenon (OR = 3.14; 95% CI 1.37 to 7.22; *p* = 0.01). In addition, Liporaci et al. [26] discovered that the concentric H:Q at 60°/s increases the risk rate for indirect injury (OR = 6.72; 95% CI 1.32–34.31; *p* = 0.02). This ratio seems to be the most promising one, as it is a strength-related risk factor that increases the likelihood of hamstring tears.

The threshold values used to define outcomes not only influence the results but are often arbitrarily chosen, representing a potential limitation of the analysis. Regarding this, Dauty et al. [38] studied the influence of setting three thresholds for diverse H:Q and H:H torque ratios. However, no significant results were obtained for any of the selected thresholds. Therefore, conducting future studies to develop objective criteria for determining risk thresholds for isokinetic strength is a priority.

With respect to the results, the strength disequilibrium of the hamstrings at 30°/s is associated with higher injury risk rates (OR = 1.05; IC 95% 1.00–1.10; *p* = 0.03); nevertheless, the magnitude of this association is small. Fousekis et al. [25] emphasize that asymmetries in hamstring eccentric strength, differences in leg length, and previous injury records are key factors for predicting the risk of muscle strain in hamstrings and quadriceps. In particular, asymmetries ≥ 15% between the eccentric strength at 60 or 180°/s of both legs’ hamstrings are associated with a significantly higher risk of tears (OR = 3.88; CI 95% 1.13–12.23); it is important to note that this achievement is supported by the good methodological quality of the study. In turn, Liporaci et al. [26] found that asymmetries ≥ 10% in the concentric torque peak at 60°/s of the quadriceps (OR = 7.49; CI 95% 1.51 to 37.26) and hamstrings (OR = 46.94; CI 95% 4.16–530) can increase the risk of hamstring tears. However, the wide confidence interval of the hamstring eccentric torque peak should be interpreted with caution.

It appears that both force asymmetry thresholds could be used as a criterion to indicate hamstring injury risk. However, based on methodological quality and confidence intervals on the magnitude of the findings, the asymmetry threshold ≥ 15% for excentric force at 60 or 180°/s of the hamstrings between limbs seems to be the most appropriate as a risk factor. Furthermore, the angular velocity of 180°/s may be more appropriate based on the speed of technical kicking actions during a soccer match [56]. Therefore, maintaining a balanced force generation between limbs seems to be essential, both in the knee extensor muscles.

### 4.3. Strength Assessment Using the Nordic Hamstring Curl Test

Hamstring strength disequilibrium, whether due to bilateral asymmetry or a disequilibrium relationship with quadriceps strength, is a modifiable risk factor associated with muscle tears in this area [57]. Hamstring injuries, common in sports like soccer [58], often occur due to excessive strain during eccentric contraction, with eccentric strength playing a crucial role in the prevention of these injuries [16]. Factors such as previous contraction speed, elongation speed, and activation time determine the severity of the injury [59]. Hamstring injuries are classified into two types according to their mechanism: those provoked by stretching, associated with movements such as kicks that combine hip flexion and knee extension, and those caused by sprints, which occur during high-speed races. While the former involves longer muscle lengths, the latter occurs within the muscle’s functional range [57].

In line with our results, Timmins et al. [42] underscore the importance of assessing knee flexor eccentric strength and torque, as they are key indicators of injuries from hamstring strains in professional soccer players. The authors identified critical thresholds based on the optimal contraction range. Specifically, the values of absolute 337 N and relative 4.35 N/kg eccentric force were determined, as well as those of absolute 145 N·m and relative 1.86 N·m/kg eccentric torque; the values were associated with higher injury risk. On the other hand, the absence of beeches by van Dyk et al. [44] could be partly explained by the quality of the methodology and the variables selected for the study since they did not consider in their analysis a displacement component, which is a crucial phenomenon in the evaluation of dynamic force. These findings emphasize the need for specific monitoring and strengthening strategies to improve the eccentric capacity of the hamstring muscles, using standardized protocols and precise tools to identify athletes at higher risk of injury. They also highlight the importance of including these strategies in preventive programs under professional supervision [60].

The relationship between hamstring strength disequilibrium and the risk of muscle tear injury remains unclear. Therefore, there is a need to explore the theoretical aspects of this phenomenon and adopt a multifactorial perspective that goes beyond strength as the sole factor [59]. Factors such as fatigue and exposure, both preseason and throughout the season, are decisive in increasing injury risk; this risk progressively rises with each match half and is reflected in the occurrence of overuse injuries [12,16]. Therefore, implementing continuous load monitoring is essential to better understand the factors that influence injury occurrence and to design more effective preventive strategies.

### 4.4. Limitations of This Study

This review has several limitations. Firstly, all included studies exclusively involved male professional soccer players, which limits the generalizability of our findings to female professional athletes. Secondly, the focus on high-level professional players means these results may not be applicable to amateur soccer players or those in sports training programs. Lastly, by concentrating on kinetic variables, this review did not consider other intrinsic factors as indicators of lower-body indirect injury risk.

Regarding the quality of the primary studies, we observed substantial methodological heterogeneity. This was particularly evident in the selection and processing of kinetic variables for both isokinetic strength assessments and the Nordic hamstring curl test. Although isokinetic evaluation protocols for knee flexor–extensors were largely consistent (e.g., contraction type and execution speed), the variables chosen for analysis varied significantly. A great diversity is observed in the ratios analyzed (based on quadriceps and hamstring strength), along with differing cut-off points for asymmetry and ratio values used to determine injury risk and even the dichotomization between dominant and non-dominant legs. Similarly, protocols and variables for the Nordic hamstring curl test differed among studies.

An important limitation concerns the use of arbitrary thresholds, such as ≥10% asymmetry or an H:Q ratio < 50%, which have not been validated through diagnostic accuracy analyses. It is recommended that future research examine these cut-off points using the receiver operating characteristic (ROC) curves and metrics such as sensitivity, specificity, and the area under the curve (AUC) in order to establish more robust and generalizable criteria. Furthermore, a key limitation identified in the primary studies is the scarcity of experimental designs, an issue that will be further addressed in the projections section.

### 4.5. Projections

Despite prevalent methodological limitations within the selected studies, this largely stems from a scarcity of experimental research in professional soccer. Implementing such designs presents considerable challenges due to potential risks for players and clubs. Consequently, prospective studies appear to be the most suitable approach for identifying indirect injury risk factors in this population. However, future research should explore the feasibility of experimental designs with amateur soccer players to more conclusively determine the relevance of clinical variables and the risk of indirect injuries.

Future research should consider other factors that may increase the risk of injury, such as age, previous injuries, and aerobic performance, among others [28,53]. For these multivariate analyses, such as machine learning algorithms (logistic regression, algorithms based on trees or neural networks) could help researchers or teams to identify variables and models that predict the probability of injury and the variables that are most important for this task [61]. Also, it should address other relevant variables, such as the rate of force development, the rate of torque development, and the integral or area under the force curve, as these metrics can provide a more comprehensive view of muscle performance under different conditions.

Although all the articles included only represented the male population, one study in the female population was identified that did not meet the selection criteria, due to the lack of formal statistical analysis between kinetic variables and injury risk [62]. Due to the growth of women’s soccer and the increased availability of resources, future studies should be conducted on a female cohort.

Another interesting point is the speed of execution of the isokinetic tests; although most of the beeches were at low speed (60°/s), this corresponds to a considerably lower speed than that achieved in technical actions such as ball kicking, where angular velocities of up to 1625°/s can be reached [56]. Therefore, it would be interesting for future studies to develop protocols with a higher contraction velocity that could have a greater ecological validity. Additionally, it would be necessary to include ecological tests involving different types of jumps performed on force platforms, as this would allow for the assessment of strength in more representative real-world performance settings. It is recommended to apply unidimensional analysis to the force’s curves using statistical parametric mapping, which could facilitate the identification of specific patterns in the dynamics of force production and help optimize training approaches [63] should also be considered relevant for a greater understanding of force patterns and their relation to injury risk.

### 4.6. Practical Applications

Implementing regular assessments at the beginning and throughout the season is key to designing comprehensive prevention and rehabilitation programs. However, conducting these analyses with caution and under the supervision of specialized professionals is fundamental to ensure proper data interpretation and the effective application of prevention strategies that benefit teams.

Permanent monitoring is essential, as well as effective communication between the interdisciplinary team, to anticipate and promptly manage injuries. Implementing these approaches will improve injury prevention and performance in professional soccer. Specifically, the maximum values for quadriceps concentric torque and hamstring eccentric torque at 60°/s, normalized by body mass, have been shown to be key predictors of hamstring tear risk. In addition, a value of less than 50.5% and greater than 64% of the H:Q concentric ratio and asymmetry ≥ 15% hamstring eccentric force at 60 or 180°/s between limbs have demonstrated a stronger association with the injury risk of this muscle group. Therefore, it is recommended that practitioners use assessment protocols that include the measurement of concentric quadriceps and hamstring strength at 60°/s, and eccentric hamstring strength at 60 and 180°/s in order to monitor indicators of hamstring injury risk without the need to subject athletes to overly extensive or intensive testing protocols.

## 5. Conclusions

Based on the results of this review, kinetic variables, particularly isokinetic torque and inter-limb asymmetries, are consistently associated with the risk of indirect musculoskeletal injuries in professional soccer players. Tests using isokinetic machines, especially at different angular velocities, have been the most reliable indicators, allowing for accurate assessments of strength deficits and muscle disequilibrium. Monitoring these variables can help identify players at higher risk, enabling the development of specific strengthening and injury prevention strategies.

This review highlights the importance of assessing muscle strength as a relevant predictor of injury risk, along with other established risk factors. For coaching and medical staff, it is essential to define risk thresholds based on these indicators, select validated tests, and apply rigorous data analysis methods. Additionally, injury risk assessment should incorporate multifactorial elements, such as well-being, fatigue, and external/internal load measures.

## Figures and Tables

**Figure 1 jfmk-10-00228-f001:**
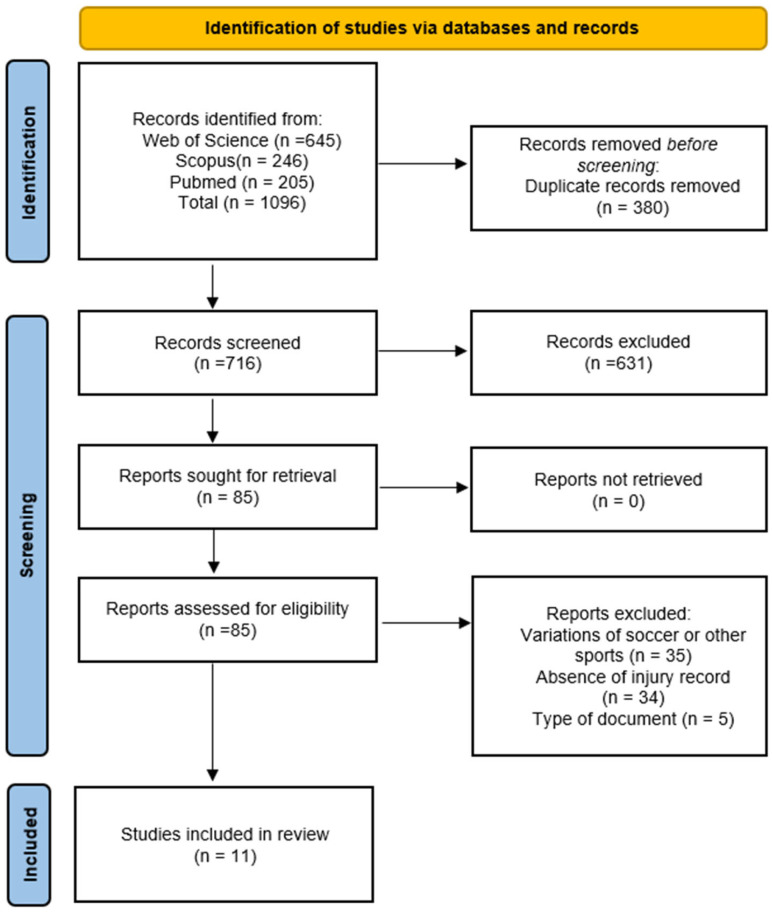
Flow diagram of item selection.

**Table 1 jfmk-10-00228-t001:** Criteria for inclusion and exclusion.

Inclusion criteria	Population: Soccer players from both sexes who are categorized as sub-elite (performance at state or national level), elite (international performance), or world-class [33].
	Intervention: Epidemiology (follow-up/history) of lower limb musculoskeletal injuries, whether these are functional direct (type 1A and 1B) or structural (type 3A, 3B, and 4) [13].
	Comparison: Dynamometric assessments of lower limb strength that are expressed in kinetic indicators such as force (N), power (W), rate of force development (N/s), impulse (N*s), and torque (N*m).
	Outcomes: The presence or absence of a relationship between kinetic indicators and indirect injuries of the lower limb as a risk factor.
**Exclusion criteria**	Conventional soccer variations such as futsal or beach soccer.
	Direct musculoskeletal injuries, bone, ligament, and joint injuries.
	Sources of information such as books, thesis, reports, and reviews.

**Table 2 jfmk-10-00228-t002:** Characteristics of population for the studies selected.

Reference	Study Design	Population				
		Sample	Mean Age (SD) Years	Mean Height (SD) m	Mean Body Mass (SD) kg	Competitive Category
Fousekis et al. [25]	Prospective cohort study	*n* = 200 16 H 84 HNI 7 Q 93 QNI	NS 22.94 (4.11) 23.00 (3.27) 25.42 (5.28) 23.42 (2.76)	NS 1.80 (4.97) 1.76 (6.19) 1.76 (4.68) 1.80 (2.76)	NS 74.71 (3.64) 71.58 (5.96) 72.35 (4.69) 73.42 (3.36)	3rd National Soccer League Division
Timmins et al. [42]	Prospective cohort study	*n* = 152 27 H 125 HNI	NS 27.0 (3.8) 24.2 (5.1)	NS 1.80 (0.07) 1.78 (0.06)	NS 76.4 (6.7) 75.3 (6.6)	Professional Australian soccer competition
van Dyk et al. [45]	Prospective cohort study	*n* = 563 167 H 396 HNI	NS 25.3 (4.9) 24.7 (4.7)	NS 1.76 (6.1) 1.76 (6.7)	NS 71.3 (7.8) 71.9 (9.1)	Qatar Stars League
van Dyk et al. [44]	Prospective cohort study	*n* = 413 66 H 347 HNI	NS 27.9 (4.3) 25.9 (4.9)	NS 1.75 (6.7) 1.76 (6.8)	NS 72.2 (7.7) 72.6 (9.2)	Qatar Stars League
Dauty et al. [38]	Prospective cohort study	*n* = 194 36 H 158 HNI	NS 23.4 (4.4) 22.4 (4.1)	NS 1.80 (4.7) 1.80 (5.6)	NS 75.4 (6) 75.1 (6.1)	French Premier and Second League
Grygorowicz et al. [40]	Retrospective cohort study	*n* = 66 11 H 55 HNI	NS	NS	NS	Polish premier league
Lee et al. [41]	Prospective cohort study	*n* = 146 41 H 105 HNI	24.2 (4.4) NS NS	1.77 (5.9) NS NS	72.9 (8.65) NS NS	Top national league
Dauty et al. [39]	Prospective cohort study	*n* = 91 31 H 60 HNI	NS 24.4 (4.4) 23.3 (4.4)	NS 1.80 (4.9) 1.80 (5.8)	NS 75.7 (6.7) 74.8 (5.9)	French Premier League
van Dyk et al. [43]	Prospective cohort study	*n* = 367 62 H 305 HNI	NS 27.8 (4.3) 26.1 (4.8)	NS 176 (7) 177 (6.8)	NS 72.3 (7.8) 72.3 (9.2)	Qatar Stars League
Liporaci et al. [26]	Prospective cohort study	*n* = 68 7 H 2 RF 4 Add	24.3 (4.6) NS NS NS	1,81 (0.07) NS NS NS	78.9 (7.76) NS NS NS	NS
Shalaj et al. [46]	Prospective cohort study	*n* = 143 40 H 103 HNI	23.2 (4.1) 26.1 (3.4) 22.2 (3.9)	180.0 (5.3) 180.5 (4.7) 179.8 (5.6)	74.2 (6.7) 77.0 (6.1) 73.2 (6.6)	Kosovo National Premier Soccer League

Note: H = hamstrings injured; HNI = hamstring not injured; Q = quadriceps injured; QNI = quadriceps not injured; RF = rectus femoral injuries; Add = adductor injuries; NS = Not specified.

**Table 3 jfmk-10-00228-t003:** Characteristics of injury follow-up for selected studies.

Reference	Injury Definition	Exposure	
		Type of Injury	Follow-Up Durations
Fousekis et al. [25]	All non-contact muscle strains force players to miss at least one scheduled practice session or game.	Hamstrings Quadriceps strain	1 season
Timmins et al. [42]	Hamstring strain injury defines any acute posterior thigh pain that resulted in the immediate cessation of exercise and is later diagnosed by the club medical staff.	Hamstrings strain	4 seasons
van Dyk et al. [45]	Hamstring strain injury is defined as acute pain in the posterior thigh that occurred during training or match play and resulted in immediate termination of play and inability to participate in the next training session or match.	Hamstrings strain	4 seasons
van Dyk et al. [44]	Hamstring strain injury is defined as acute pain in the posterior thigh that occurred during training or match play and resulted in immediate termination of play and inability to participate in the next training session or match.	Hamstrings strain	2 seasons
Dauty et al. [38]	Physical complaint in the region of the posterior thigh sustained during a soccer match or training, without contact, irrespective of the need for medical attention, with time loss from soccer activities (7 to 28 days).	Hamstrings strain	1 season
Grygorowicz et al. [40]	The player was unable to take part in a match or in a training session due to hamstring strain that happened in a football match or during training, and at least one of the following consequences was present: decrease in the quantity or level of sports activity for at least one day, or need or medical evaluation or non-operative or operative treatment.	Hamstrings strain	1 Season
Lee et al. [41]	Hamstring strain injury is defined as acute pain in the posterior thigh, which causes immediate cessation of match play or training.	Hamstrings strain	1 season
Dauty et al. [39]	Moderate injuries: time loss between 7 and 28 days. Major injuries: time loss > 28 days.	Hamstrings strain	1 season
van Dyk et al. [43]	Hamstring injury is defined as acute pain in the posterior thigh that occurred during training or match play and resulted in immediate termination of play and inability to participate in the next training session or match.	Hamstrings strain	2 seasons
Liporaci et al. [26]	Any physical complaints resulting in a player being unable to take part in at least one subsequent football training session or match.	Hamstrings strain Rectus femoris strain Adductor strain	1 season
Shalaj et al. [46]	Not specified	Hamstrings strain	2 seasons

**Table 4 jfmk-10-00228-t004:** Methodology quality assessment. Modified Downs and Black checklist [34].

Authors	Reporting										External Validity			Internal Validity							Selection Bias						Power	Total	Quality
	1	2	3	4	5	6	7	8	9	10	11	12	13	14	15	16	17	18	19	20	21	22	23	24	25	26	27		
Fousekis et al. [25]	1	1	1	1	2	1	1	0	1	1	0	0	0	0	0	1	1	1	1	1	1	1	0	0	1	1	1	20	Good
Dauty et al. [38]	1	1	1	1	0	1	1	0	1	1	0	0	0	0	0	1	1	1	1	1	1	1	0	0	0	1	0	16	Fair
Dauty et al. [39]	1	1	1	1	0	1	1	0	1	1	0	0	0	0	0	1	1	1	1	1	1	1	0	0	0	1	0	16	Fair
Grygorowicz et al. [40]		1	1	1	0	1	1	0	1	1	0	0	0	0	0	1	1	1	1	1	1	0	0	0	0	1	0	15	Fair
Lee et al. [41]	1	1	0	1	1	1	1	0	1	1	0	0	0	0	0	1	1	1	1	1	1	1	0	0	1	1	1	18	Fair
Liporaci et al. [26]	1	1	1	1	1	1	1	0	1	1	0	0	0	0	0	1	1	1	1	1	1	1	0	0	0	1	0	17	Fair
Shalaj et al. [46]	1	1	1	1	1	1	1	0	1	1	0	0	0	0	0	1	1	1	1	1	1	1	0	0	1	1	1	19	Fair
Timmins et al. [42]	1	1	1	1	2	1	1	0	1	1	0	0	0	0	0	1	1	1	1	1	1	1	0	0	1	1	1	20	Good
van Dyk et al. [45]	1	1	1	1	1	1	1	0	0	1	1	0	0	0	0	1	0	1	1	1	1	0	0	0	1	0	0	15	Fair
van Dyk et al. [44]	1	1	1	1	1	1	1	0	1	1	1	1	0	0	0	1	1	1	1	1	1	0	0	0	0	0	1	18	Fair
van Dyk et al. [43]	1	1	1	1	1	1	1	0	1	1	1	1	0	0	0	1	0	1	1	1	1	0	0	0	0	0	0	16	Fair

**Table 5 jfmk-10-00228-t005:** Glossary.

	Definition
Absolute maximum torque	Maximum torque generated by a joint during a voluntary maximal contraction without considering the individual’s body weight [47].
Relative maximum torque	Maximum torque normalized to body weight (Nm/kg), allowing for comparisons between individuals of different sizes [47].
Asymmetry between limbs	Difference in strength or functional performance between the right and left limbs [48].
H:Q ratio	Ratio between the strength of the hamstrings and quadriceps muscles. Used to assess muscular balance at the knee joint and prevent injury [49].
Dynamic control ratio	Functional comparison between eccentric torque of hamstrings and concentric torque of quadriceps, which is important for dynamic knee stability [50].
Strength imbalance	Significant strength difference between opposing muscles (agonists vs. antagonists) or between limbs, potentially leading to compensatory movement patterns and increased injury risk [51].
Rate of torque development	Speed at which torque is generated from the onset of a muscular contraction. Calculated as the slope (first derivate over time) of the torque–time curve during isometric contraction [47].

Only two studies [25,26] based their analyses on dichotomous variables for interlimb asymmetry and the H:Q ratio. Fousekis et al. [25] defined interlimb asymmetry as a difference of ≥15% and strength asymmetry in the H:Q ratio as a value < 1. Conversely, Liporaci et al. [26] used a threshold of ≥10% for interlimb asymmetry and considered an H:Q ratio outside the 55–64% range to indicate strength asymmetry.

**Table 6 jfmk-10-00228-t006:** Strength evaluation by isokinetic dynamometry.

References	Kinetics Variables	Outcome of Analysis Univariate	Outcome of Analysis Multivariate
Fousekis et al. [25]	Asymmetry ≥ 15% to Q con strength (yes/no) Asymmetry ≥ 15% to Q ecc strength (yes/no) Asymmetry ≥ 15% to H con strength (yes/no) Asymmetry ≥ 15% to H ecc strength (yes/no) Asymmetry < 1 to ratio H ecc 180°/s:Q con 180°/s (yes/no)	Eccentric hamstring asymmetries (OR = 3.88; CI al 95% 1.13–12.23; *p* = 0.03)	Not applicable
van Dyk et al. [45]	H con at 60°/s and 300°/s absolute (Nm) H con at 60°/s and 300°/s adjusted (Nm/kg) H ecc at 60°/s absolute (Nm) H ecc at 60°/s adjusted (Nm/kg) Q con at 60°/s and 300°/s absolute (Nm) Q con at 60°/s and 300°/s adjusted (Nm/kg) Ratio Q:H con 60°/s (AU) Ratio Q:H con 300°/s (AU) Ratio Q con 300°/s: H ecc 60°/s (AU)	Not applicable	Quadriceps concentric 60°/s adjusted (OR = 1.41; 95% CI 1.03 to 1.92; *p* = 0.03) Hamstrings eccentric 60°/s adjusted (OR = 1.37; 95% CI 1.01 to 1.85; *p* = 0.04)
van Dyk et al. [44]	H con 60°/s and 300°/s absolute (Nm) H con 60°/s and 300°/s adjusted (Nm/kg) H ecc 60°/s absolute (Nm) H ecc 60°/s adjusted (Nm/kg) Q con 60°/s and 300°/s absolute (Nm) Q con 60°/s and 300°/s adjusted (Nm/kg) Ratio H ecc 60°/s: Q con 300°/s (UA) Dynamic control ratio (UA) Dynamic control ratio con at 30°, 40° and 50° (UA) Dynamic control ratio ecc at 30°, 40° and 50° (UA) Overall H:Q ratio (UA)	Categorical variable (criterion: >1 SD above the mean) Q Concentric at 300°/s adjusted (HR = 2.06; IC 95% 1.21 to 3.51; *p* = 0.008)	Not applicable
Dauty et al. [38]	H:H con 60°/s R/L < 0.9; ≤0.85; <0.87 H:H con 60°/s L/R < 0.9; <0.85; <0.86 H:H ecc 30°/s R/L < 0.9; <0.85; <0.80 H:H ecc 30°/s L/R < 0.9; <0.85; <0.83 H:Q con 60°/s R < 0.6; ≤0.47; <0.55 H:Q con 60°/s L < 0.6; ≤0.47; <0.55 Hecc30°/s/Qcon240°/s R < 1; <0.80; <1.01 Hecc30°/s/Qcon240°/s L < 1; <0.80; <0.99	No significative results	Not applicable
Grygorowicz et al. [40]	Q con 60°/s absolute (Nm) Q con 60°/s relative (Nm/kg) H con 60°/s absolute (Nm) H con 60°/s relative (Nm/kg) Ratio H:Q con 60°/s (AU) Ratio H:Q con 60°/s Cut-off score < 0.47 (AU) Ratio H:Q con 60°/s Cut-off score < 0.6 (AU) Ratio H:Q con 60°/s Cut-off score < 0.658 (AU)	No significative results	Not applicable
Lee et al. [41]	H con 60°/s and 240°/s absolute (Nm) H con 60°/s and 240°/s relative (Nm/kg) H ecc 30°/s absolute (Nm) H ecc 30°/s relative (Nm/kg) Q con 60°/s and 240°/s absolute (Nm) Q con 60°/s and 240°/s relative (Nm/kg) Ratio H:Q con 60°/s (%) Ratio H:Q con 240°/s (%) Ratio H ecc 30°/s: Q con 240°/s (%) Strength imbalance H con 60°/s (Nm) Strength imbalance H con 240°/s (Nm) Strength imbalance H con 30°/s (Nm)	H strength absolute value of Con 60°/s (OD = 0.97; 95% CI 0.95 to 0.99; *p* = 0.002) H strength relative value of Con 60°/s (OD = 0.60; 95% CI 0.14 to 0.26; *p* < 0.001) H strength absolute value of Con 240°/s (OR = 0.97; 95% CI 0.94 to 1.00; *p* = 0.03) H strength relative value of Con 240°/s (OR = 0.36; 95% CI 0.00 to 0.30; *p* = 0.03) H strength absolute value of Ecc 30°/s (OR = 0.98; 95% CI 0.97 to 0.99; *p* = 0.002) H strength relative value of Ecc 30°/s (OR = 0.15; 95% CI 0.06 to 0.40; *p* < 0.001) H/Q ratio con 60°/s (OR = 0.92; 95% CI 0.87 to 0.97; *p* = 0.001) H strength imbalance concentric at 30°/s (OR = 1.05; 95% CI 1.00 to 1.10; *p* = 0.03)	Preseason hamstring strength measures at 30 deg/s, Nm/kg ≤ 2.40 (adjusted OR = 5.59; 95% CI 2.20 to 12.92; *p* < 0.001) Preseason hamstring to quadriceps ratios Con 60/Con 60 (%) ≤ 50.5 (adjusted odd ratio = 3.14; 95% CI 1.37 to 7.22; *p* = 0.01)
Dauty et al. [39]	Ratio H:H con 60°/s (AU) Ratio H:H ecc 30°/s (AU) Ratio Q:Q con 60°/s (AU) Ratio Q:Q con 240°/s (AU) Ratio H:Q con 60°/s (AU) Ratio H ecc 30°/s: Q con 240°/s (AU)	H/Hcon60°/s (Ndom/dom): OR = 38; IC 95% 1.06 to 1818; *p* = 0.04	Not applicable
van Dyk et al. [43]	H RTD 30 ms con 60°/s (Nm/s) H RTD 50 ms con 60°/s (Nm/s) H RTD 100 ms con 60°/s (Nm/s) H RTD 30 ms con 300°/s (Nm/s) H RTD 50 ms con 300°/s (Nm/s) H RTD 100 ms con 300°/s (Nm/s) Q RTD 30 ms ecc 60°/s (Nm/s) Q RTD50 ms ecc 60°/s (Nm/s) Q RTD 100 ms ecc 60°/s (Nm/s)	No significative results	Not applicable
Liporaci et al. [26]	Asymmetry > 10% of knee flexion 60°/s (Yes/no). Asymmetry > 10% of knee extension 60°/s (Yes/no). Ratio H:Q 60°/s between 55 and 64% (Yes/no)	Ext PT 10 (OR = 7.49; CI (95%) 1.51–37.26; *p* = 0.01) Flex PT 10 (OR = 46.94; CI (95 %) 4.16–530; *p* < 0.01) H:Q ratio con 60°/s (OR = 6.72; CI (95 %) 1.32–34.31; *p* = 0.02)	Not applicable
Shalaj et al. [46]	H dominant con 60°/s and 240°/s absolute (Nm) H non-dominant con 60°/s and 240°/s absolute (Nm) Q dominant con 60°/s and 240°/s absolute (Nm) Q non-dominant con 60°/s and 240°/s absolute (Nm) H dominant ecc 30°/s and 120°/s absolute (Nm) H non-dominant ecc 30°/s and 120°/s absolute (Nm) H dominant con 60°/s and 240°/s relative (Nm/kg) H non-dominant con 60°/s and 240°/s relative (Nm/kg) Q dominant con 60°/s and 240°/s relative (Nm/kg) Q non-dominant con 60°/s and 240°/s relative (Nm/kg) H dominant ecc 30°/s and 120°/s relative (Nm/kg) H non-dominant ecc 30°/s and 120°/s Relative (Nm/kg) Ratio H/Q con 60°/s dominant absolute (%) Ratio H/Q con 60°/s non-dominant absolute (%) Ratio H/Q con 240°/s dominant absolute (%) Ratio H/Q con 240°/s non-dominant absolute (%)	Not applicable	Concentric hamstring 240°/s dominant leg (β = 0.01 ± 0.01; CI 0.00 to 0.01; *p* = 0.049)

Notes: con = concentric; ecc = eccentric; H = hamstring; Q = quadriceps; RTD = rate of torque development; OR = Odds ratio; HR = Hazard Ratio; dom = dominant; Ndom = non-dominant.

**Table 7 jfmk-10-00228-t007:** Strength evaluation by Nordic hamstring curl.

References	Kinetics Variables	Outcome of Analysis Univariate	Outcome of Analysis Multivariate
Timmins et al. [42]	Eccentric force (N) Eccentric torque (Nm) Relative eccentric force (N/Kg) Relative eccentric torque (Nm/Kg) Isometric force (N) Isometric torque (N/m) Relative isometric force (N/Kg) Relative isometric torque (Nm/Kg)	Eccentric force ROC-curve determined threshold of 337 N (RR = 4.4; 95% CI 1.1 to 17.6; *p* = 0.013) Eccentric torque ROC-curve determined threshold of 145 N/m (RR = 3.6; 95% CI 1.2 to 11.4; *p* = 0.017) Relative eccentric force ROC-curve determined threshold of 4.35 N/kg (RR = 2.5; 95% CI 1.1 to 6.2; *p* = 0.041) Relative eccentric torque ROC-curve determined threshold of 1.86 Nm/kg (RR = 2.9; 95% CI 1.1 to 7.1; *p* = 0.011)	Model 3: Mean eccentric strength of both limbs (N) (X^2^ = 6.33; *p* = 0.011) Model 4: Mean eccentric strength of both limbs (N) (X^2^ = 5.05; *p* = 0.024) Model 5: Mean eccentric strength of both limbs (N) (X^2^ = 4.29; *p* = 0.038)
van Dyk et al. [44]	Peak force absolute (N) Peak force adjusted (N/kg) Peak force imbalance absolute (N) Peak force imbalance adjusted (N/kg) Average force absolute (N) Average force adjusted (N/kg)	No significative results	Not applicable

Note: ROC = receiver operating characteristic; RR = relative risk.

## Data Availability

The original contributions presented in this study are included in the article. Further inquiries can be directed to the corresponding author.

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
