# Peer review of "Kinetic Variables as Indicators of Lower Limb Indirect Injury Risk in Professional Soccer: A Systematic Review"

_jfmk, 2025, doi:10.3390/jfmk10020228_

Round 1

Reviewer 1 Report

Comments and Suggestions for Authors

The article "Kinetic Variables as Indicators of Lower Limb Indirect Injury Risk in Professional Soccer: A Systematic Review" presents a solid systematic review, but it contains several areas that require substantial corrections, clarifications, and refinements that could significantly enhance its scientific value and practical applicability.

1. Lack of inclusion of female population
All included studies were limited to male participants. While the authors briefly mention this in the discussion, it was not addressed in the search strategy (e.g., lack of additional keywords like “female” or “women”). This should be treated as a critical limitation, and the authors should explicitly recommend future research involving female athletes.

2. Too general methodological quality analysis
The study quality assessment is based on the Downs and Black scale, but the authors do not specify which particular domains were most often rated poorly. There is no stratified analysis of how methodological quality influenced the findings—e.g., whether studies rated “good” produced more consistent results than “fair” ones. This weakens the conclusions, as it is unclear how reliable the data are. The inclusion of a quality-of-evidence grading system like GRADE would strengthen the review.

3. Methodological heterogeneity without attempts to resolve it
There is significant heterogeneity in measurement methods (e.g., different angular speeds, devices, protocols, normalization approaches). The authors acknowledge this issue but make no attempt to group studies by methodology or to control for this variability. A subgroup categorization table or more structured analysis would help clarify the findings.

4. Incomplete statistical synthesis
The article lacks a meta-analysis, which might be justified due to data heterogeneity, but the authors do not attempt even basic approaches to harmonize results (e.g., converting ORs or HRs to a common metric). A narrative synthesis with effect direction and consistency (e.g., vote counting or a forest plot without meta-analysis) would have added clarity.

5. Arbitrary thresholds used as risk cut-offs without diagnostic evaluation
Reported thresholds (e.g., ≥10% asymmetry, H:Q ratio <50%) are presented as risk indicators, but their diagnostic accuracy is not analyzed—no ROC curves, sensitivity/specificity, or AUCs are discussed. Often, these thresholds are adopted from single studies and lack cross-validation, which undermines their generalizability.

6. Undefined or insufficiently explained terminology
Some technical terms (e.g., “dynamic control ratio,” “RTD,” “torque asymmetry”) are not adequately defined, which may confuse readers unfamiliar with biomechanical testing. A glossary of terms or expanded explanations in the methods section would improve accessibility.

7. Lack of discussion on confounding variables and co-factors
Although the paper discusses “modifiable” risk factors, there is no analysis of how factors like injury history, training load, age, or playing position influence the observed relationships. These should have been controlled for in multivariate analysis or at least discussed as potential confounders.

8. Limited practical recommendations
Although the authors mention the need for strength monitoring, the practical implications are vague. They should provide specific actionable recommendations, such as: “if H:Q < X, implement intervention Y during period Z,” ideally based on empirical findings. Monitoring protocols should also be proposed.

9. Inconsistencies in data presentation and referencing
Some tables (e.g., with strength test results) are detailed but inconsistent in formatting. Demographic data are incomplete in some cases. Standardizing table layouts and simplifying abbreviations would enhance clarity and professionalism.

10. Language bias not discussed
The authors included studies in English and Spanish. However, they do not explain the exclusion of other languages like German, Portuguese, or French—languages commonly present in databases such as Scopus or Web of Science. This limitation should be acknowledged.

In summary, while the article provides a valuable and well-structured review on injury prevention in professional soccer, it needs improvements in standardizing methodologies, refining the analysis of study quality, and offering more detailed and practical guidance. Addressing these areas would significantly enhance the article’s scientific robustness and clinical utility.

Author Response

Dear Reviewer 1

All responses have been included in the attached file. All changes are indicated in red in the text.

Regards

Reviewer 2 Report

Comments and Suggestions for Authors
  1. Title and Abstract

Suggestions:

  • Title clarity: Consider simplifying the title to make it more accessible, e.g., “Kinetic Variables as Predictors of Lower Limb Injuries in Team Sports: A Systematic Review.”
  • Abstract clarity: The abstract lacks specific information about the main findings. Add 1–2 sentences summarizing the key conclusions (e.g., which kinetic variables were most relevant).
  • Keywords: Add keywords after the abstract to enhance searchability (e.g., “biomechanics,” “injury prevention,” “ground reaction force,” etc.).
  1. Introduction

Strengths:

  • Provides a relevant and timely topic.
  • Shows awareness of injury epidemiology and screening limitations.

Revisions:

  • Problem statement: Clarify the gap this review addresses—e.g., “Although kinetic variables are used in performance contexts, their utility in injury prediction remains unclear.”
  • Hypothesis or aim: Be more explicit. Consider a final paragraph that ends with: “This review aimed to systematically examine the role of kinetic variables in predicting non-contact lower limb injuries in team sports.”
  1. Methods

Revisions:

  • Search strategy: Include the search strings used in databases (PubMed, Scopus, Web of Science).
  • Inclusion/exclusion criteria: Consider presenting them in a table for readability.
  • Quality assessment: The modified Downs and Black checklist is appropriate, but state how many reviewers did the scoring and how discrepancies were resolved.
  • PRISMA diagram: Include a complete one in the appendix if not already submitted.
  1. Results

Revisions:

  • Summary of studies: Include a comprehensive Table 1 summarizing each study: author/year, sample size, sport, kinetic variable(s), injury definition, follow-up period, primary outcomes.
  • Findings structure: Break down into sub-sections (e.g., “Vertical Ground Reaction Force,” “Lateral Imbalance,” “Propulsion Forces”) to guide the reader more effectively.
  • Statistical trends: Where possible, report effect sizes, p-values, or trends even if insignificant, to show patterns across studies.
  1. Discussion

Strengths:

  • Integrates findings with broader literature and acknowledges multifactorial injury mechanisms.

Revisions:

  • Clarity: Clarify whether any variable showed strong evidence across multiple studies (e.g., was high vertical force at landing consistently linked to injury?).
  • Practical applications: Be more specific—what should coaches/trainers do differently based on these findings?
  • Limitations: Add more detail—e.g., high heterogeneity in study design, injury definitions, and follow-up durations.
  • Future directions: Suggest specific future study designs (e.g., prospective cohort studies with standardized force plate protocols).
  1. Conclusion

Revisions:

  • Be more assertive about the main takeaway.
  • Avoid vague phrasing like “there is an important need to investigate further…”—say what specifically should be investigated.
  1. Style and Language

General Suggestions:

  • Review grammar and flow—some sentences are long or awkwardly structured.
  • Define abbreviations (e.g., “GRF” should be defined on first use).
  • Please ensure consistency in tense: use the past tense when describing previous studies.
  1. Figures and Tables

Suggestions:

  • Add a summary table of kinetic variables and their relationship with injury (e.g., ↑ GRF = ↑ injury risk in 4/6 studies).
  • Could you include illustrative figures where appropriate (e.g., a flowchart of kinetic screening vs injury monitoring)?
  1. References

Revisions:

  • Ensure references are up-to-date and include recent studies from 2022–2024 if available.
  • Use a consistent citation style (APA, Vancouver, etc.) and double-check for formatting errors.
Comments on the Quality of English Language

The English could be improved.

Author Response

Dear Reviewer 2

All responses have been included in the attached file. All changes are indicated in red in the text.

Regards

Reviewer 3 Report

Comments and Suggestions for Authors

Thanks for submitting the manuscript. The study was well designed and implemented. There are some minor issues need to be addressed by the authors:

1) On p. 3, the authors may further clarify on which measures they are going to focus on this study, in particular the one related to kinetic variables and musculoskeletal injuries.

2) Please provide more justifications for just focusing the period from 01/01/2010 to 14/11/2023 (p. 4).

3) When reporting the findings, please clarify why the Cochrane reviews format was not used to show the figures of the study.

Author Response

Dear Reviewer 3

All responses have been included in the attached file. All changes are indicated in red in the text.

Regards
